# The Role of Transforming Growth Factor-Beta in Retinal Ganglion Cells with Hyperglycemia and Oxidative Stress

**DOI:** 10.3390/ijms21186482

**Published:** 2020-09-04

**Authors:** Hsin-Yi Chen, Yi-Jung Ho, Hsiu-Chuan Chou, En-Chi Liao, Yi-Ting Tsai, Yu-Shan Wei, Li-Hsun Lin, Meng-Wei Lin, Yi-Shiuan Wang, Mei-Lan Ko, Hong-Lin Chan

**Affiliations:** 1Institute of Bioinformatics and Structural Biology & Department of Medical Sciences, National Tsing Hua University, Hsinchu 300, Taiwan; dful690@gmail.com (H.-Y.C.); nakyla1215@gmail.com (E.-C.L.); peach0722@hotmail.com.tw (Y.-T.T.); t91050127@hotmail.com.tw (Y.-S.W.); lishlin0207@gmail.com (L.-H.L.); eva1018cat@yahoo.com.tw (M.-W.L.); woowoow0320@gmail.com (Y.-S.W.); 2Department of Ophthalmology, National Taiwan University Hospital Hsin-Chu Branch, Hsinchu 300, Taiwan; phify66@gmail.com; 3Department of Biomedical Engineering and Environmental Sciences, National Tsing Hua University, Hsinchu 300, Taiwan; chouhc@mail.nhcue.edu.tw

**Keywords:** TGF-β, retinal ganglion cells, hyperglycemia, oxidative stress

## Abstract

A characteristic of diabetes mellitus is hyperglycemia, which is considered with an emphasis on the diabetic retinopathy of progressive neurodegenerative disease. Retinal ganglion cells (RGCs) are believed to be important cells affected in the pathogenesis of diabetic retinopathy. Transforming growth factor-beta (TGF-β) is a neuroprotective protein that helps to withstand various neuronal injuries. To investigate the potential roles and regulatory mechanisms of TGF-β in hyperglycemia-triggered damage of RGCs in vitro, we established RGCs in 5.5, 25, 50, and 100 mM D-glucose supplemented media and focused on the TGF-β-related oxidative stress pathway in combination with hydrogen peroxide (H_2_O_2_). Functional experiments showed that TGF-β1/2 protein expression was upregulated in RGCs with hyperglycemia. The knockdown of TGF-β enhanced the accumulation of reactive oxygen species (ROS), inhibited the cell proliferation rate, and reduced glutathione content in hyperglycemia. Furthermore, the results showed that the TGF-β-mediated enhancement of antioxidant signaling was correlated with the activation of stress response proteins and the antioxidant pathway, such as aldehyde dehydrogenase 3A1 (ALDH3A1), heme oxygenase-1 (HO-1), nuclear factor erythroid 2-related factor (Nrf2), and hypoxia-inducible factor (HIF-1α). Summarizing, our results demonstrated that TGF-β keeps RGCs from hyperglycemia-triggered harm by promoting the activation of the antioxidant pathway, suggesting a potential anti-diabetic therapy for the treatment of diabetic retinopathy.

## 1. Introduction

Diabetic retinopathy can cause vision loss and blindness. The underlying cause leading to retinopathy remains unclear; however, the study has implicated that hypoxia, ischemia, oxidative stress, and metabolic mechanisms are involved in the retinal microvascular pathogenesis, which may also induce inflammation and neuroglial degeneration [1,2,3,4,5,6]. Hyperglycemia is believed to play a crucial role in the pathogenesis of retinal microvascular damage. In general, high glucose promotes the non-enzymatic modification of the tissue proteins; the transformation and dysfunction of proteins resulting from the formation of advanced glycation end products (AGEs) weaken the retinal capillaries, cause the leakage of blood into the surrounding space, and play a central role in the pathogenesis of diabetic complications [7]. The four major mechanisms involved in diabetic retinopathy are increased hexosamine pathway flux, increased AGE formation, and abnormal activation of the protein kinase C (PKC) pathway and neurodegeneration pathways. The AGEs contribute to the formation of crosslinks and barriers between the extracellular matrix and receptor for advanced glycation end products. AGEs binding to cell receptors leads to the accumulation of reactive oxygen species (ROS) in the retina and other eye tissues [8]. ROS activate many signal transduction pathways such as the activation of inflammation or protein kinase and phosphorylation. The mechanisms of ROS-induced cell damage results in the modification of proteins and changes their functions. The free thiol group (RSH) of cysteine residues of proteins are a dominant nucleophilic agent that is capable of carrying out a series of redox reactions in hyperglycemic conditions, leading to protein abnormality. Moreover, excess glucose-triggered oxidative stress can accelerate the development of diabetic complications and cause RGCs damage. Recent studies [9,10] have shown that an excessive accumulation of ROS is an intrinsic mechanism associated with the pathogenesis of diabetes-associated microvascular complications such as diabetic nephropathy. By the modification of specific phospholipid and glycolipid species in the intermediate oxidative steps, hyperglycemia-triggered oxidative stress resulting in the overproduction of mitochondrial ROS has been implicated in these retinopathic mechanisms [11,12]. Studies have demonstrated that increased mitochondrial dysfunction and cell death was caused by high glucose treatment [13,14]. In addition, ROS generation is significantly increased in the diabetic mouse retina, while the suppression of ROS generation effectively inhibited visual impairment and caspase-3-mediated retinal neuronal apoptosis [15].

The hyperglycemia-triggered dysregulation of various pathways affected by oxidative stress either promote the apoptosis of damaged cell or cellular dysfunction. Thus, oxidative stress has become a crucial influencing factor in the development of retinopathy in patients with diabetes [16]. Hyperglycemia promotes the formation of AGEs, initiates oxidative impairments, and increases hypoxia-inducible factor (HIF)-1 alpha protein levels, which lead to the upregulation of VEGF and transactivation function [17]. Nrf2 is also an important regulator of oxidative-stress-related protein, including heme oxygenase-1 [18], and it plays a central role in the protection against oxidative and apoptotic damage in diabetic retinopathy [19]. Recent studies show that the renal protective effects in diabetic rats might be enhanced by the activation of Nrf2 [20,21]. The activation of Nrf2 is a stimulating initiator for the induction of neuroprotection pathways, and dissociation of the cytoplasmic Keap1 and Nrf2 complex represents a signal transduction of the cellular mechanisms for oxidative stress [22].

Transforming growth factor-beta (TGF-β) is a group of structurally related, multifunctional regulatory proteins, which have pleiotropic functions in various organs. They regulate cellular processes in normal physiology including the proliferation, recognition, differentiation, and survival of different cell types and thus can be considered to be growth factors [23]. The TGF-β family includes three isoforms (TGF-β1, TGF-β2, and TGF-β3) [24]. These three isoforms of TGF-β subtypes are encoded by separate genes but demonstrate homology in protein sequence and have similar mechanisms for initiation and activation [25]. It has been indicated that TGF-β has an anti-oxidative cytoprotective effect against cell damage in specific cell types. Nevertheless, the involvement of TGF-β in the hyperglycemia-triggered oxidative damage of RGCs remains unknown. We aimed to investigate the functional status of RGCs in hyperglycemia-triggered oxidative damage. We further studied the potential roles of TGF-β regulated mechanisms in correlated pathways of signal transduction in case of oxidative damage, as well as, in case of the combined effects of hydrogen peroxide and hyperglycemia-induced oxidative injury. Our study will help to establish the mechanisms of pathology and survival in the treatment of glaucoma, diabetes retinopathy, and diabetic retinopathy in a glaucoma state.

## 2. Results

### 2.1. Effects of Hyperglycemia on RGCs

To explore the effects of hyperglycemia on RGC-5 cells, glucose was supplemented to the medium in different concentrations (5.5, 25, 50, and 100 mM glucose), and proliferation assays were carried out on RGC-5 cell cultures following 3 weeks at the above-mentioned specific glucose concentrations. As shown in Figure 1A, RGC-5 cells seeded in 96-well plates were subjected to four different glucose concentrations and observed for four days. Glucose concentrations decreased the cell proliferation rate in a concentration-dependent manner. Cells with higher glucose concentration (100 mM) in medium showed a significantly lower proliferation rate. The cell proliferation rates in 5.5 mM and 25 mM glucose supplementation reached 7- and 10-fold (normalized to Day 1), respectively, while 50 mM and 100 mM glucose supplemented cells only proliferated to 5- and 3-fold, respectively. In order to investigate whether hyperglycemic treatment of RGC-5 cells accumulated intracellular ROS, ROS were detected by aminophenyl fluorescein (APF) and flow cytometry using Accuri Cflow and Cflow Plus analysis software (BD Biosciences, San Jose, CA, USA), which was used to analyze ROS accumulation by the altered signal percentages compared to the peak of the control group (5.5 mM). This is represented in Figure 1B. Concentrations of 50 mM and 100 mM glucose increased the percentages of peak shift, to 4.2% and 18.2%, respectively. Thus, the state of hyperglycemia was shown to directly affect RGC-5 cell ROS production in a concentration-dependent manner. iROS (intracellular reactive oxygen species) accumulation was increased in the RGCs under oxidative stress induced by high glucose concentration. To study whether hyperglycemia affects mitochondria activity, we conducted an oxygen consumption rate (OCR) analysis by high-resolution respirometry using an Oxygraph-2K platform. In Figure 1C, our results demonstrated that cells cultured in 100 mM glucose concentrations medium have a lower reserve respiration capacity and membrane integrity, as compared to the control group (5.5 mM), implying that hyperglycemic status may impact mitochondrial activity, especially in terms of reserve respiration capacity. We measured the GSH (Glutathione) content in RGCs with hyperglycemia and investigated changes of GSH content in RGC-5 cells with different glucose concentrations (5.5, 25, 50, and 100 mM). In 100 mM-glucose treated RGC-5 cells, we found that higher glucose concentration in medium reduced cellular GSH content. The higher the glucose concentration in the growth medium, the greater the observed decrease of GSH levels. This result suggested that our conditions of hyperglycemic state altered the GSH redox environment in RGCs (Figure 1D).

### 2.2. Immunoblot Analysis of RGCs with Hyperglycemia

To identify the potential role of signal transductions in regulating hyperglycemia-induced oxidative damage in RGCs, we cultured RGCs in medium supplemented with different concentrations of glucose for at least 3 weeks to establish long-term hyperglycemia status and then detected changes in the expression of Nrf2, Keap1 (Kelch-like ECH-associated protein 1), HIF-1α, ALDH3A1, HO-1, TGF-β1, and TGF-β2 proteins in response to hyperglycemia. High glucose concentration induces alterations in protein expression, and our results (Figure 2) showed increased protein expression in RGCs, indicating that antioxidation/neuroprotective signaling pathways may play critical roles in regulating the hyperglycemic state in RGCs.

### 2.3. Immunofluorescence Analysis of RGCs with Hyperglycemia

To investigate if high glucose concentration-induced oxidative stress may cause retinal ganglion cell fibrosis, we performed the immunofluorescence analysis to examine the fibers and actin dots in RGCs. We observed that more stress fibers and actin dots formed in RGCs cultured in 50 mM and 100 mM glucose compared with cells cultured in 5.5 mM and 25 mM glucose, which is consistent with previous data of the dose-dependency (Figure 3). In addition, TGF-β1 and TGF-β2 knockdown promotes more thick stress fibers and actin dots formation than the Ctrl group in each glucose concentration group.

### 2.4. Effects in TGF-β1/2 Knockdown RGCs with Hyperglycemia

To demonstrate whether TGF-β exerts a protective effect against high glucose concentration-triggered oxidative damage, we knocked down TGF-β in cells and cultured them in 5.5, 25, 50, and 100 mM glucose supplementation for at least 3 weeks, thereby detecting the changes in the proliferation rate of TGF-β knockdown on hyperglycemia-induced oxidative damage. Our results (Figure 4A) showed that the knockdown of TGF-β1 resulted in a significant decrease in cell proliferation both in low and high glucose concentrations. Hyperglycemia treatment-induced inhibition in cell proliferation of RGCs was intensified by TGF-β knockdown. These results suggested that TGF-β protects RGCs against high glucose concentration-induced oxidative damage and that TGF-β1/2 knockdown aggravates hyperglycemia-induced oxidative damage in RGCs. Subsequently, we detected the iROS accumulation of TGF-β knockdown RGCs with hyperglycemia. Our results (Figure 4B) showed that the ROS accumulation in hyperglycemia-treated RGCs was markedly accelerated by the TGF-β1 knockdown. In 25 mM glucose concentration, knockdown mediated by shTGF-β1 and shTGF-β2 increased the peak shift percentages from 21.2% to 31.2% and 25.8%, respectively. In 50 mM glucose concentration, shTGF-β1 and shTGF-β2 increased the peak shift percentages from 22.4% to 35.3% and 25.5%, respectively. In 100 mM glucose concentration, shTGF-β1 and shTGF-β2 increased the peak shift percentages from 26.6% to 35.5% and 28.2%, respectively. As mentioned above, these results suggested that TGF-β protects RGCs against hyperglycemia-triggered oxidative damage and that TGF-β1 knockdown aggravates hyperglycemia-induced oxidative damage in RGCs. Meanwhile, we investigated the changes of GSH content in TGF-β1/2 knockdown RGCs incubated in 5.5, 25, 50, and 100 mM Dulbecco’s modified Eagle’s medium (DMEM). We found that the shTGF-β1 and shTGF-β2 treated groups showed reduced cellular GSH content in each glucose concentration, especially the shTGF-β1 group. These results also showed that TGF-β1 knockdown aggravates hyperglycemia-induced oxidative damage in RGCs (Figure 4C).

### 2.5. Immunoblot Analysis of Activation of Downstream Substrates in TGF-β1/2 Knockdown RGC-5 Cells Treated with Hyperglycemia

As shown in Figure 5, to establish TGF-β-related pathway collusion with high glucose concentration, we cultured RGCs in different glucose concentrations containing medium for at least 3 weeks to establish long-term hyperglycemia status and further detected changes in the expression of Nrf2, Keap1, HIF-1α, ALDH3A1, and HO-1 in response to oxidative stress. The protein expression of Nrf2, Keap1, ALDH3A1, and HO-1, which are related to the self-protection of cells and antioxidation, were downregulated with TGF-β1/2 knockdown. Overall, these results suggested that TGF-β induces downstream signal transduction, which can shield RGCs against hyperglycemia-triggered oxidative damage.

### 2.6. Effects in RGCs with Hyperglycemia with or w/o rhTGF-β1 Protein (5 ng/mL)

To explore whether TGF-β promotes an anti-oxidative pathway against high glucose concentration-triggered oxidative damage, we established experiments with rhTGF-β1 protein (5 ng/mL) followed by detecting the effects of TGF-β upregulation on hyperglycemia-triggered oxidative damage. Our results (Figure 6A) showed that the upregulation of TGF-β1 resulted in a significant increase in cell proliferation of high glucose concentration-treated RGCs. Especially, the increased cell proliferation rate of the 100 mM glucose concentration-treated group was markedly reversed to an approximate rate of 5.5 mM by TGF-β overexpression. Overall, these results suggested that TGF-β1 protected RGCs against hyperglycemia-induced oxidative damage and promoted cell proliferation. Subsequently, we detected the iROS accumulation of RGC-5 cells with hyperglycemia with or *w*/*o* recombinant TGF-β1 protein (5 ng/mL). Our results (Figure 6B) showed that the ROS production in hyperglycemia-treated RGCs declined by TGF-β1 overexpression. In 50 mM glucose concentration, the group with rhTGF-β1 reduced the peak shift percentages from 27.6% to 15.3%. In 100 mM glucose concentration, the group with rhTGF-β1 reduced the peak shift percentages from 33.2% to 22.4%. In summary, these results suggested that TGF-β1 protects RGCs against hyperglycemia-triggered oxidative damage and that TGF-β1 knockdown aggravates hyperglycemia-induced oxidative damage in RGCs. Similarly, we investigated the GSH content changes in RGCs incubated in 5.5, 25, 50, and 100 mM with or without recombinant TGF-β1 protein (5 ng/mL). We found that the groups with rhTGF-β1 (5 ng/mL) added show the higher cellular GSH content in each glucose concentration, especially in 50 mM and 100 mM. These results also showed that TGF-β1 maintains the content of glutathione under the conditions of hyperglycemia-induced oxidative damage in RGCs (Figure 6C).

### 2.7. Immunoblot Analysis in RGCs with Hyperglycemia with or w/o Recombinant TGF-β1 Protein (5 ng/mL)

In Figure 7, to further study the TGF-β-related pathway involvement with high glucose concentration, we first cultured RGCs in different glucose concentrations medium for at least 3 weeks to establish long-term hyperglycemia status, further treated them with recombinant TGF-β1 protein (5 ng/mL), and detected changes in the protein expression of Nrf2, Keap1, HIF-1α, ALDH3A1, and HO-1. The expression of Nrf2, Keap1, ALDH3A1, and HO-1 in the rhTGF-β1 group, related to self-protection in cells and antioxidation, were upregulated with TGF-β1 overexpression. Overall, these results suggested that TGF-β1 induces downstream signal transduction, which can switch on antioxidation pathway in RGCs.

### 2.8. Effects in RGCs with Hyperglycemia with or without Hydrogen Peroxide for 1 h

To explore the additional effects in hyperglycemia- and H_2_O_2_-induced oxidative damage, we performed experiments with these two oxidative stress induction factors and subsequently detected the cell viability post induction of oxidative damages. Accordingly, the concentration of glucose was shown to affect the tolerance to hydrogen peroxide in a concentration-dependent manner (Figure 8A). The higher the glucose concentration, the greater the sensitivity to H_2_O_2_-induced oxidative stress. The IC_50_ concentration of hydrogen peroxide in 100 mM glucose concentration was much lower than that in 5.5 mM and 25 mM glucose concentration. Likewise, we detected the iROS accumulation in RGC-5 cells with hyperglycemia with or without hydrogen peroxide (1 mM) for 1 h. Our results (Figure 8B) showed that the ROS production increased in hyperglycemic condition followed by H_2_O_2_ treatment. Overall, these results suggested that hyperglycemia- and H_2_O_2_-induced oxidative damage can produce more iROS in RGCs, which implies that patients with diabetic retinopathy may suffer more serious retina damage when combined with glaucoma. In addition, we investigated the GSH content changes in RGCs incubated in 5.5, 25, 50, and 100 mM glucose supplemented medium with or without hydrogen peroxide (1 mM) for 1 h. We found that the groups with hydrogen peroxide treatment show the lower cellular GSH content in each glucose concentration, especially in 50 mM and 100 mM in Figure 8C. These results also showed the addition reaction in RGCs.

### 2.9. Immunoblot Analysis in RGCs with Hyperglycemia with or without Hydrogen Peroxide (1 mM) for 1 h

As shown in Figure 9, to further study the TGF-β-related pathway involvement with high glucose concentration, we first cultured RGCs in different glucose concentrations containing medium for at least 3 weeks to establish long-term hyperglycemia status, further treated them with H_2_O_2_ (1 mM) for 1 h, and subsequently detected changes in protein expression of Nrf2, Keap1, HIF-1α, ALDH3A1, HO-1, and TGF-β1/2. The expression of Nrf2, Keap1, ALDH3A1, HO-1, and TGF-β1/2, which is related to the self-protection of cells and antioxidation, was upregulated with H_2_O_2_ (1 mM) treatment for 1 h and showed additional oxidative effects.

### 2.10. A Hypothetical Model Detailing the Role of TGF-β in Hyperglycemia

In a hyperglycemic state, the accumulated ROS promotes TGF-β conversion from latent form to active form to induce the downstream signaling. Upregulated TGF-β in RGCs subsequently induces Nrf2, Keap1, ALDH3A1, and HO-1. The activation of these accelerates more TGF-β expression, which facilitates the antioxidant pathways to induce neuroprotection. On the other hand, increased HIF-1α may damage mitochondria function and promote fibrosis and actin dots formation. Thus, TGF-β1 protects RGCs against hyperglycemia-induced oxidative damage by facilitating cell antioxidation and neuroprotection pathways (Figure 10).

## 3. Discussion

A study on the damage of retinal ganglion cells is essential for a better understanding of the development of diabetic retinopathy and the progression of the disease. In the previous research, we indicated the important role of TGF-β1/2 in regulating oxidative stress-induced damage in RGCs and showed that TGF-β1/2 knockdown promoted apoptosis and ROS accumulation in RGCs induced by H_2_O_2_ treatment. However, the oxidative stress induced by H_2_O_2_ treatment mimics the state of glaucoma. Therefore, we hypothesize that a similar pathway also contributes to oxidative stress in the retina by hyperglycemia, which may cause retinal ganglion cell dysfunction and apoptosis. In this research, we summarized an important role of TGF-β1/2 in limiting hyperglycemia-induced damage in RGCs. Our data showed that RGCs with hyperglycemia inhibited cell proliferation and promoted ROS production, which induced oxidative stress in cells. We illuminated that the molecular signal transduction underlying the TGF-β1-mediated protective effect is associated with its promotion of expression of Nrf2, Keap1, HIF-1α, ALDH3A1, and HO-1 proteins, suggesting that TGF-β1 may act as a potential initiator for retinal protection in diabetic retinopathy.

Li et al. [26] reported that the levels of TGF-β1 are significantly elevated not only in renal biopsy specimens derived from diabetic nephropathy (DN) patients, but they are also elevated in various renal cells, including podocytes, mesangial cells, and proximal tubular epithelial cells. This evidence shows that TGF-β1 could play an important role in the pathogenesis of diabetic retinopathy. Liu et al. [27] indicated that there are increased levels of CTGF (connective tissue growth factor), VEGF (Vascular endothelial growth factor), and TGF-β2 expression and apoptosis in the rat retina early in diabetes, and the degree of increase becomes greater as diabetes develops. Tsin et al. [28] showed that the benefits of the TGF-β pathway blockade can induced cell viability and reverse hyperglycemia-induced cytokine overproduction. Additionally, TGF-β2 is involved in the pathogenesis of proliferative vitreoretinal diseases [29]. Conversely, More et al. [30] showed the anti-inflammatory characteristic of TGF-β in the neuroprotective action in Parkinson’s disease. Dobolyi et al. [31] also described the potential mechanisms of anti-oxidative effects exerted by TGF-β including anti-inflammatory, anti-apoptotic, and anti-cytotoxic actions as well as the promotion of scar formation, angiogenesis, and neuroprotection in the review paper, and also declared TGF-βs to be a potentially continuous trophic support factor for many cell types. Before binding to its receptors (serine/threonine kinases, TGF-βR), TGF-β, when produced, are bound to the latency-associated peptide (LAP), rendering them inactive, while being activated in the target tissue by enzymes that cleave LAP, and then active TGF-β can bind to its receptors [32]. Based on these studies, we demonstrated that TGF-β1/2 protein expressions were increased in RGCs after hyperglycemia or high glucose concentration treatment, and the overexpression of TGF-β1 suppressed high glucose-induced ROS production with enhanced cell functions.

Nevertheless, the RGC-5 cell line, which is a non-human cell line originally thought to be derived from rat retinal ganglion cells, has been criticized for its characteristics recently [33]. RGC-5 cells were identified to be of mouse origin, and their expression of RGC characteristics was questioned by some laboratories. Yet, considering the in vitro experiments, we must deeply think of the many manual and realistic aspects. The primary cultured retinal ganglion cells are unable to stably survive for a period of time, and before the in vivo validation, the stable RGC-5 cell line has been widely used as a cell culture model to study the cell physiology of RGCs. Ultimately, our purpose is to investigate the effects of oxidative stress and its possible signal transduction in the cellular level. On the other hand, when we discuss the data of the group of 5.5 mM and 25 mM, we can find that the data in the 25 mM group shows better cell function and capacity or viability. This is the reason why 25 mM is the glucose concentration of normal cell line maintaining medium. This also explains why the data from 5.5 mM glucose supplementation is usually not the perfect condition in cell functional assay.

Using proliferation assay and ROS reagent combined with flow cytometry Accuri Cflow and Cflow Plus analysis software and other cell functional assays, we provided strong evidence that RGCs in culture with high glucose concentration medium showed a significantly lower proliferation rate, lower reserve respiration capacity and membrane integrity, reduced cellular GSH content, and increased iROS. In addition, the stress fiber and actin dots formation that enhance cell fibrosis were consistently detected in these hyperglycemic conditions.

As pointed out in our previous study, TGF-β mediates the individual processes of development, growth, inflammation, immune regulation, and oxidative stress response [34]. Our results showed that TGF-β1 seems more effective than TGF-β2 in the anti-oxidative protection system. TGF-β1/2 knockdown RGCs showed lower proliferation and GSH content, while higher iROS were produced in each glucose concentration. In immunoblot analysis of the cellular anti-oxidation pathway, we found that the expression of neuroprotective protein factors such as Nrf2, ALDH3A1, and HO-1 decreased, indicating the crucial role of TGF-β1/2 in regulating the switch to anti-oxidative signal transduction. Accordingly, the group with recombinant TGF-β1 protein (5 ng/mL) representing the effect of TGF-β1 overexpression demonstrated higher proliferation and GSH content, while lower iROS were produced in each glucose concentration. Interestingly, shTGF-β1 and shTGF-β2 treated groups showed reduced cellular GSH content in each glucose concentration, especially at low glucose concentrations; meanwhile, the groups with rhTGF-β1 (5 ng/mL) added show the higher cellular GSH content in each glucose concentration, especially in 50 mM and 100 mM. This may explain why TGF-β acts as a dual-functional protein, which is capable of protecting against or accelerating injury; the effects of TGF-β for neuroprotection may be limited. When we combined the two oxidative stress-inducible factors, hyperglycemia and hydrogen peroxide, the data revealed that the tolerance to H_2_O_2_ in high glucose concentration became lower than that of the control group; these data implied that in patients with diabetes combined with glaucoma state, the damage to RGCs is more serious than those whose plasma glucose levels are in control. The accumulation of AGEs caused by hyperglycemia elevate the levels of intracellular ROS and cause irreversible cell damage through epigenetic changes such as histone modifications, DNA methylation, and non-coding RNAs [35,36]. However, additional experiments will be needed to further clarify the significance of AGEs and the hyperglycemia-induced signal transduction.

Summarizing the results, our findings demonstrated that TGF-β1 protected RGCs against hyperglycemia-induced oxidative damage by promoting cell antioxidation and neuroprotection pathways, including Nrf2/Keap1/ALDH3A1/HO-1 signaling (Figure 10). These findings implied that TGF-β1 may play important roles in the antioxidation system of RGCs. Considering that antioxidant signal transduction is a promising therapeutic target in retinopathy diseases, targeting TGF-β1 to promote Nrf2/Keap1/ALDH3A1/HO-1 signaling may be a candidate therapeutic approach for diabetic retinopathy. This would help to better understand the underlying molecular mechanisms that regulate disease progression, potential treatment modalities involving these antioxidative benefits, and guide the selection of therapeutic targets. Meanwhile, this study will also help to establish the pathomechanisms and survival treatment of glaucoma, diabetes, and diabetes in a glaucoma state, respectively.

## 4. Materials and Methods

### 4.1. Cell Line and Cell Culture

The transformed RGC cell line, RGC-5 was a gift from Dr. Yang in National Taiwan University Hospital, Taiwan. RGC-5 cells were maintained in Dulbecco’s modified Eagle’s medium (DMEM) supplemented with 10% fetal bovine serum and 1% antibiotic–antimycotic (all from Gibco Invitrogen Corp., Paisley, UK). All cells were incubated at 37 °C in 5% CO_2_. For the condition of different glucose concentrations, cells were exposed to D-glucose at a final concentration of 5.5, 25, 50, and 100 mM glucose concentration (the corresponding blood glucose level is 100, 454, 910, and 1600 mg/dL, respectively). Glucose concentrations of 25, 50, and 100 mM corresponded to plasma glucose levels 2 h post-meal in diabetic patients and glucose levels in uncontrolled diabetic patients [37] and were compared with cultures exposed to 5.5 mM D-glucose as control (corresponding to fasting plasma glucose levels of diabetes-free control) [17,38,39]. To avoid the hyperosmotic stress in high glucose concentration, mannitol was used to balance the osmolarity in differential glucose concentrations according to a previous report [40]. After exposure for at least 20 days, the long-term hyperglycemic state monolayer cultured cells were used for further analysis.

### 4.2. Proliferation Assay

Proliferation rates were estimated using MTT (3-(4,5-Dimethylthiazol-2yl)-2,5-diphenyltetrazolium bromide) solution (USB Corp., Cleveland, OH, USA). The cells were seeded into 96-well plates at a density of 2 × 10^3^ cells each well. After a 24 h adhesion (Day 1), the media was removed, and the cells were incubated in 100 μL of MTT solution (1 mg/mL) per well for 3.5 h at 37 °C in the dark. The supernatant was then removed, and 100 μL of dimethyl sulfoxide (DMSO) was added in each well. After the plates were shaken for 30 s to completely dissolve the insoluble purple formazan, the absorbance at 570 nm was measured by a plate reader. The cell growth rate measurement for each group was estimated by a similar method on Day 2, Day 3, and Day 4. The proliferation rates were shown as a ratio normalized to Day 1.

### 4.3. Cell Viability Assay

The cell viability rate was investigated using MTT solution (USB Corp., Cleveland, OH, USA). Cells were seeded into 96-well plates at a density of 8 × 10^3^ cells per well. The media was discarded after 24 h of incubation, and the cells were induced with different concentrations and exposure durations of hydrogen peroxide for 1 h.The hydrogen peroxide-contained media was removed, and cells were then further incubated in 100 μL per well of 1XMTT solution (1 mg/mL) for 3.5 h at 37 °C in the dark. Then, the MTT solution was replaced with 100 μL per well of dimethyl sulfoxide (DMSO) was added. The 96-well plates were shaken for 30 s before detection to completely dissolve the insoluble purple formazan, and the absorbance of each well was measured at 570 nm.

### 4.4. Detection of ROS

Aminophenyl fluorescein (APF; Goryo Chemical, Sapporo, Japan) was used to quantify ROS generation. First, 1 mg of APF powder was dissolved in 0.47 mL of N, N-dimethylformamide (DMF) to prepare the solution with a final concentration of 5 mM. The reagent was diluted to 5 μM by PBS (Phosphate buffered saline) or serum-free medium before adding to the cells. After the addition of APF, the cells were incubated in dark for 15 min, and then the fluorescence emission was measured at 515 nm post excitation at 490 nm. The extent of APF fluorescence was quantified by using Accuri CFlow@ and CFlow Plus software (BD Biosciences) [41]. The fluorescence shift percentage was normalized to the Ctrl (5.5 mM) group.

### 4.5. Immunoblotting Analysis via Western Blot

The quantified protein samples were separated in a 12% gel and transferred to polyvinylidene difluoride (PVDF) membranes (Pall Corp., Port Washington, NY, USA). After the membranes were blocked with 5% (*w*/*v*) skimmed milk or BSA (bovine serum albumin) in Tris-buffered saline with Tween-20 [TBST; 50 mM Tris, 150 mM NaCl, and 0.1% Tween-20 (*v*/*v*); pH 8.0] for 1 h, the primary antibodies (Genetex Inc., Hsinchu, Taiwan, Table 1) were diluted 1:2000 and added onto the membranes, which were incubated overnight to 48 h at 4 °C. Following this, the membranes were washed in TBST for 6 times (10 min/wash) and then incubated in TBST solution containing appropriate horseradish peroxidase-coupled secondary antibodies (Jackson ImmunoResearch Laboratories, Inc., West Grove, PA, USA) in dilution of 1:10,000 for 1 h with gentle agitation. Then, the membranes were washed again in TBST 6 times (10 min/time), and the probed proteins were visualized using an enhanced chemiluminescence (ECL) method (Visual Protein Biotech Corp., Taipei, Taiwan) [42]. The data of densitometric quantification based on ratio to housekeeping protein (LDH) were evaluated by Image J software.

### 4.6. Immunofluorescence

The cells were seeded at a density of 3 × 10^4^ cells each well onto 12 mm-coverslips (VWR International Corp., Radnor, PA, USA) plated in 24 wells and incubated overnight for further experiments. Then, the cells were fixed with PBS containing 4% (*v*/*v*) paraformaldehyde (PFA) for 25 min at 37 °C and washed three times with PBS. The coverslips were incubated in PBS containing 0.1% (*v*/*v*) Triton X-100 not longer than 5 min to permeabilize the cell membrane. Coverslips were washed and blocked in PBS containing 5% (*w*/*v*) BSA for 1 h before incubation with Phalloidin, which is tagged with fluorescent dyes selectively binding to F-actin; then, it is diluted in 2.5% BSA/PBS at the ratio 1:400 for the optimized duration. Then, coverslips were washed three times with PBS and were sealed to glass slides with Prolong^TM^ Antifade Mounting Medium (Thermo Fisher Scientific, Waltham, MA, USA) containing DAPI and then stored overnight at 4 °C in the dark. For analysis, cells were imaged using a Zeiss Axiovert 200 M fluorescent microscope (Carl Zeiss, Oberkochen, Germany). Images were exported as.zvi files using Zeiss Axioversion 4.0 and processed using Adobe Photoshop V.7.0 software.

### 4.7. Treatment of Cells with Recombinant Human TGF-β Protein1 (rhTGF-β1)

The cells were maintained in serum-free medium for 24 h before being exposed to a 24-h treatment with 5 ng/mL rhTGF-β1 (Sino Biological, Běijīng, China) protein or were left untreated in the control group. Whole-cell extracts were prepared in the NP40 cell lysis buffer for further immunoblotting analysis or ROS detection or MTT assay.

### 4.8. Measurement of Intracellular GSH Content

ApoGSH^TM^ Glutathione Detection Kit (BioVision Corp., San Francisco, CA, USA) with spectrofluorimetry was used to analyze intracellular GSH content. The cells were pooled into and centrifuged at 1000× *g* for 5 min to remove the supernatant. Rapidly, 100 μL 5% sulfosalicylic acid was added to the cell pellets followed by vortexing vigorously and incubating on ice for 10 min, which was followed by centrifugation at 12,000× *g* for 20 min at 4 °C. The supernatants were transferred into new tubes for glutathione assays and kept on ice. The samples were diluted 10-fold with GSH Assay Buffer. The diluted samples were added to the 96-well plate and to the background control.Meanwhile, the GSH standard curve was also added to the 96-well plate. The Reaction Mix was added to each well containing the GSH standard, and the Sample Background Control Mix was added to the wells containing sample background controls. Absorbance was measured in a fluorescence plate reader at OD = 450 nm at room temperature for 0, 20, 40, and 60 min, respectively. Results were calculated and expressed as ng/mL of sample [43].

### 4.9. High-Resolution Respirometry

Oxygraph-2K (Oroboros, Innsbruck, Austria) instrument and DatLab software were applied to determine cellular oxygen consumption rates. The density of RGC-5 cells was 1 × 10^6^/mL in a total volume of 2.5 mL. Oligomycin/triethyltin bromide (TET, 2 mM), carbonilcyanide p-triflouromethoxyphenylhydrazone (FCCP, 4 mM), 2 mM Rotenone and 2 mM antimycin A were added during the process. Mitochondrial basal respiration (R), ATP production, proton leakage (L), and maximal respiration (E) were evaluated by determining oxygen consumption rates in the presence of these mitochondrial inhibitors. To compare mitochondrial functionality with different basal respiration activity (routine oxygen consumption), oxygen consumption rates were converted to comparable values, which were calculated (E-R)/E as the reserve capacity [44].

### 4.10. Statistical Analysis

Data and figures are plotted as means ± standard error of the mean (SEM). Differences between the experimental groups were assessed using paired Student’s t-test, or by one & two-way analysis of variance (ANOVA). Test results with *p* < 0.05 were considered statistically significant. (*, *p* < 0.05; **, *p* < 0.01; and ***, *p* < 0.001).

## Figures and Tables

**Figure 1 ijms-21-06482-f001:**
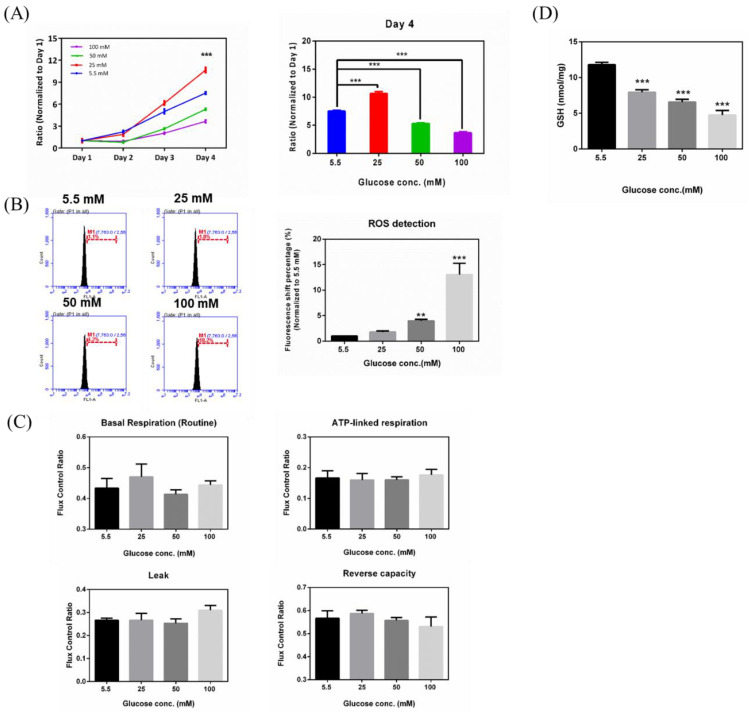
Effects of hyperglycemia on retinal ganglion cells (RGCs). (**A**) Left: Cell proliferation rates under different glucose concentrations were examined from Day 1 to Day 4 by cell viabilityassay. The data from Day 2, Day 3, and Day 4 were normalized to Day 1. Right: Isolated Day 4 data to show the differential proliferation rates in four glucose concentrations. Data are represented as mean ± SEM. ***, *p* < 0.001 when compared to the 5.5 mM group. (**B**) Left: Increased iROS (intracellular reactive oxygen species) accumulation in RGCs with hyperglycemia. Intracellular reactive oxygen species (ROS) contents were determined by the aminophenyl fluorescein (APF) assay on RGC-5 cell cultures at different glucose concentrations (5.5 mM, 25 mM, 50 mM, and 100 mM glucose) for the duration of 3 weeks. After exposure, 106 cells were stained with diluted APF (5 μM). Intracellular ROS generation was quantified by flow cytometry. Right: Relative fluorescence shift percentage was quantified and normalized to the 5.5 mM group. Data are represented as mean ± SEM. **, *p* < 0.01; and ***, *p* < 0.001 when compared to 5.5 mM group. (*n* = 3) (**C**) The oxygen consumption rate (OCR) of RGCs with hyperglycemia in order of Basal respiration (routine), ATP-linked respiration, Leak, and Reverse capacity. Data are represented as means ± SEM. (*n* = 3) (**D**) Measurement of GSH content in RGCs with hyperglycemia. Dose responses involving total GSH (Glutathione) levels in cells treated with 5.5 mM, 25 mM, 50 mM, and 100 mM glucose medium were assayed. Data are represented as means ± SEM. ***, *p* < 0.001 when compared to the 5.5 mM group. (*n* = 3).

**Figure 2 ijms-21-06482-f002:**
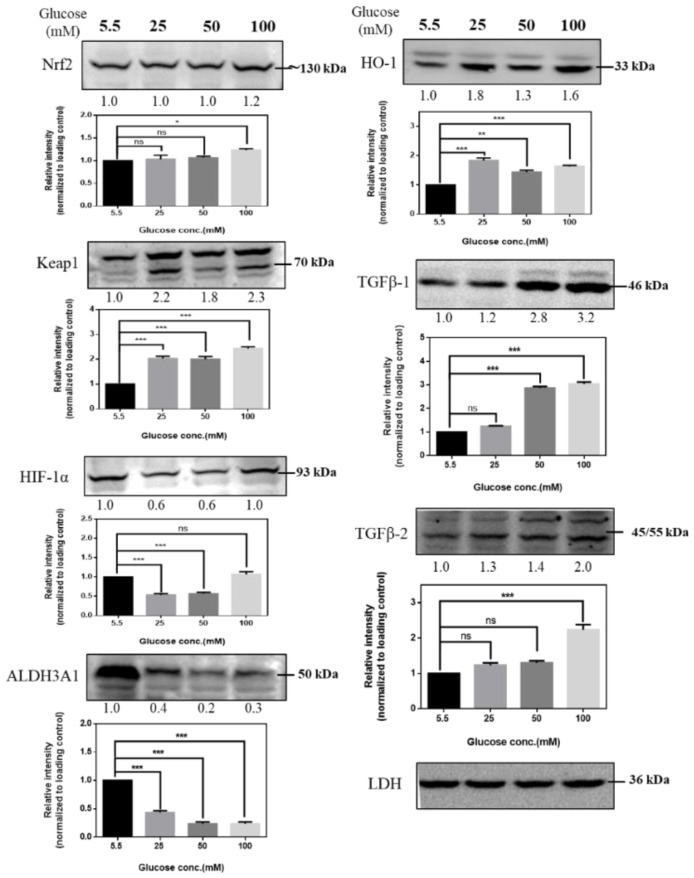
Immunoblot analysis of RGCs with hyperglycemia. Immunoblot analysis of antioxidation pathway-associated proteins (Nrf2, Keap1, HIF-1α, ALDH3A1, HO-1, TGF-β1, TGF-β2) under 5.5 mM, 25 mM, 50 mM, and 100 mM glucose medium. The protein expression values were quantified in relation to the loading control. Data are represented as means ± SEM. *, *p* < 0.05; **, *p* < 0.01; ***, *p* < 0.001; ns, nonsignificant, when compared to the 5.5 mM group (*n* = 3). ALDH3A1: aldehyde dehydrogenase 3A1, HIF-1: hypoxia-inducible factor, HO-1: heme oxygenase-1, Nrf2: nuclear factor erythroid 2-related factor, Keap1: Kelch-like ECH-associated protein 1, TGF-β: Transforming growth factor-beta.

**Figure 3 ijms-21-06482-f003:**
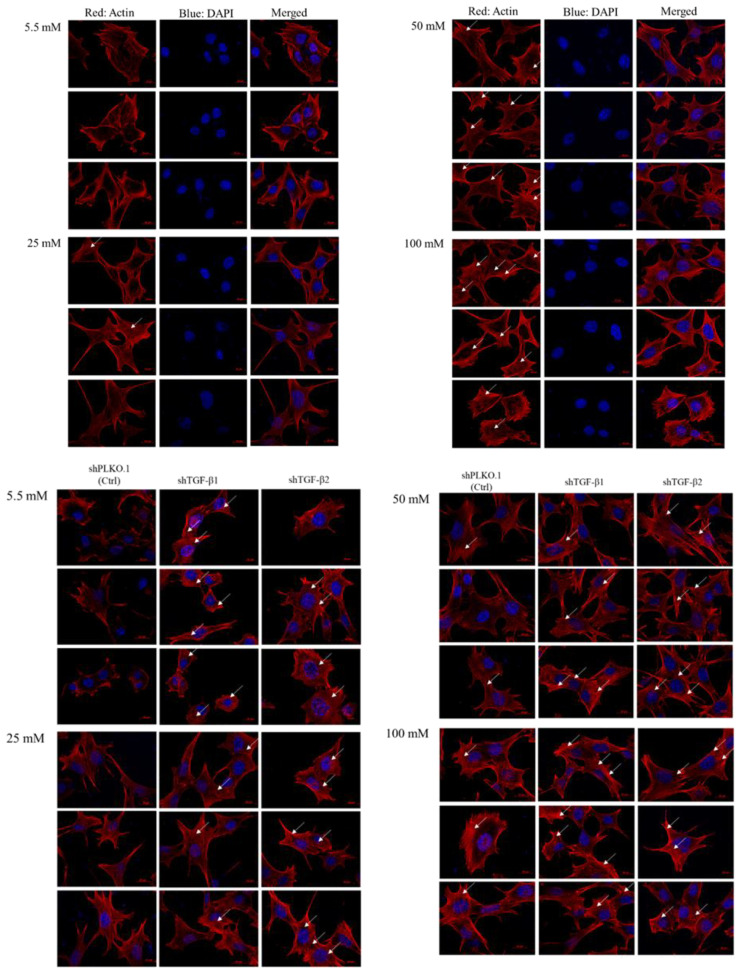
Immunofluorescence analysis of the activation of stress fiber and actin dots formation with 5.5 mM, 25 mM, 50 mM, and 100 mM glucose supplemented medium. RGC-5 and shPLKO.1 and shTGF-β1 and shTGF-β2 cells incubated in 5.5 mM, 25 mM, 50 mM and 100 mM glucose supplemented medium for at least 3 weeks on 12 mm-coverslips were fixed and stained with Phalloidin and DAPI (4′,6-diamidino-2-phenylindole). Each set of fields was taken using the same exposure, and images are representative of 3 different fields. (Scale bar = 20 μm).

**Figure 4 ijms-21-06482-f004:**
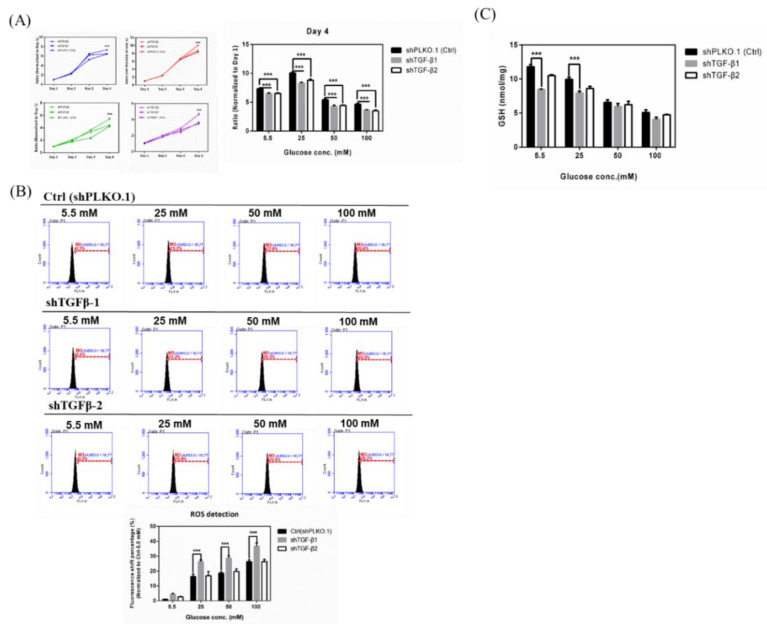
Effects in TGF-β1/2 knockdown RGCs with hyperglycemia. (**A**) Left: Cell proliferation rates under different glucose concentrations were observed from Day 1 to Day 4 by cell viability assay. The data from Day 2, Day 3, and Day 4 were normalized to Day 1. TGF-β1/2 knockdown RGC-5 cells were incubated in 5.5 mM, 25 mM, 50 mM, and 100 mM glucose for 3 weeks or longer and plated into 96-well plates to observe the proliferation rate in different glucose concentrations. Right: Isolated Day 4 data show the differential proliferation rate in TGF-β1/2 knockdown in RGC-5 cells in four different glucose concentrations. Data are represented as mean ± SEM. ***, *p* < 0.001 when compared to shPLKO.1 (Ctrl). (**B**) Upper: iROS production in TGF-β1/2 knockdown RGCs with hyperglycemia. Intracellular ROS accumulations were assessed by the APF detection method in TGF-β1/2 knockdown RGC-5 cell cultures in long-term different glucose concentrations (5.5 mM, 25 mM, 50 mM, and 100 mM glucose). After exposure, 106 cells were treated with APF (5 μM) in PBS (phosphate buffered saline). Intracellular ROS generation was quantified by flow cytometry. (*n* = 3) Lower: Relative fluorescence shift percentage was quantified and normalized with respect to the shPLKO.1 (5.5 mM) group. Data are represented as mean ± SEM. ***, *p* < 0.001 when compared to shPLKO.1 (Ctrl). (**C**) GSH content in TGF-β1/2 knockdown RGCs with hyperglycemia. Dose responses involving total GSH levels in TGF-β1/2 knockdown cells treated with 5.5 mM, 25 mM, 50 mM, and 100 mM glucose medium were assayed. Data are represented as means ± SEM. ***, *p* < 0.001 when compared to the shPLKO.1 (Ctrl).

**Figure 5 ijms-21-06482-f005:**
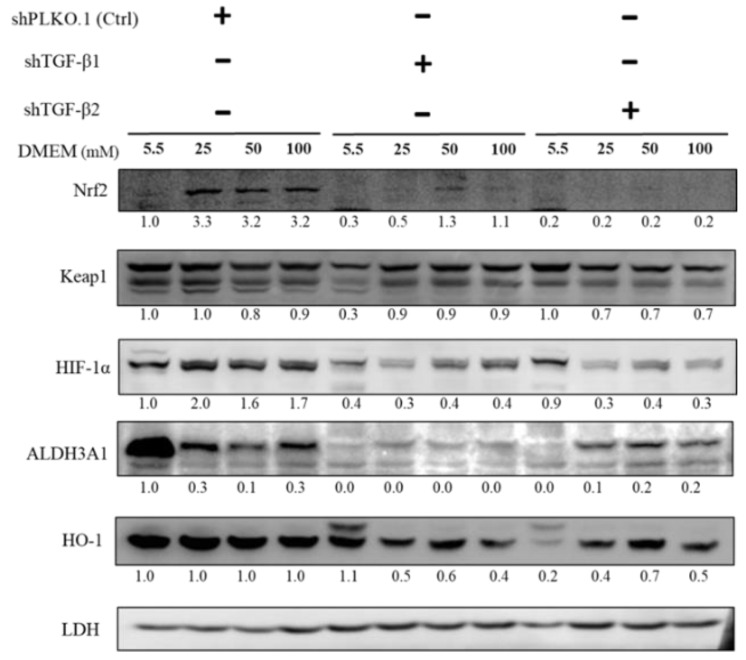
Immunoblot analysis of activation of downstream substrates in TGF-β1/2 knockdown RGC-5 cells with hyperglycemia. Immunoblot analysis of antioxidation pathway-associated proteins (Nrf2, Keap1, HIF-1α, ALDH3A1, and HO-1) in 5.5 mM, 25 mM, 50 mM, and 100 mM glucose supplemented medium. The protein expression values were quantified in relation to the loading control. Data are represented as means ± SEM. **, *p* < 0.01; ***, *p* < 0.001; ns, non-significant, when compared to the 5.5 mM group (*n* = 3).

**Figure 6 ijms-21-06482-f006:**
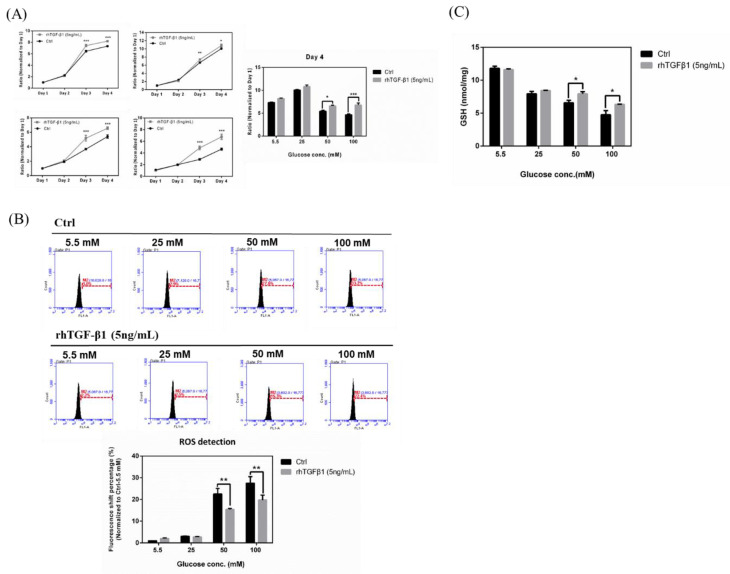
Effects in RGCs with hyperglycemia with or without (*w*/*o*) recombinant TGF-β1 protein (5 ng/mL). (**A**) Left: Cell proliferation rates under different glucose concentrations were examined from Day 1 to Day 4 using the cell viability assay. The data from Day 2, Day 3, and Day 4 were normalized to Day 1. RGC-5 cells were incubated in 5.5 mM, 25 mM, 50 mM, and 100 mM glucose for long-term exposure and plated into 96-well plates to observe the proliferation rate with or *w*/*o* recombinant TGF-β1 protein (5 ng/mL) in different glucose concentrations. Right: Isolated Day 4 data show the differential proliferation rate in RGC-5 cells with recombinant TGF-β1 protein in four glucose concentrations. Data are represented as mean ± SEM. *, *p* < 0.05; and ***, *p* < 0.001 when compared to Ctrl. (**B**) Upper: iROS accumulation effects in RGCs with hyperglycemia with or *w*/*o* recombinant TGF-β1 protein (5 ng/mL). Intracellular ROS content were evaluated by the APF staining method on RGC-5 cell cultures following 3 weeks at different glucose concentrations (5.5 mM, 25 mM, 50 mM, and 100 mM glucose) with or *w*/*o* recombinant TGF-β1 protein (5 ng/mL). After exposure, 106 cells were incubated with APF at a specific concentration (5 μM) in PBS. Intracellular ROS generation was quantified by flow cytometry. (*n* = 3) Lower: Relative fluorescence shift percentage was quantified and normalized with respect to the Ctrl (5.5 mM). Data derived from three independent experiments are presented as mean ± SEM. **, *p* < 0.01 when compared to Ctrl. (**C**) GSH content in RGCs with hyperglycemia with or *w*/*o* recombinant TGF-β1 protein (5 ng/mL). Dose responses involving total GSH contents in RGC-5 cells treated with 5.5 mM, 25 mM, 50 mM and 100 mM glucose concentration medium with or *w*/*o* recombinant TGF-β1 protein were assayed. Data are represented as means ± SEM. *, *p* < 0.05 when compared to the Ctrl.

**Figure 7 ijms-21-06482-f007:**
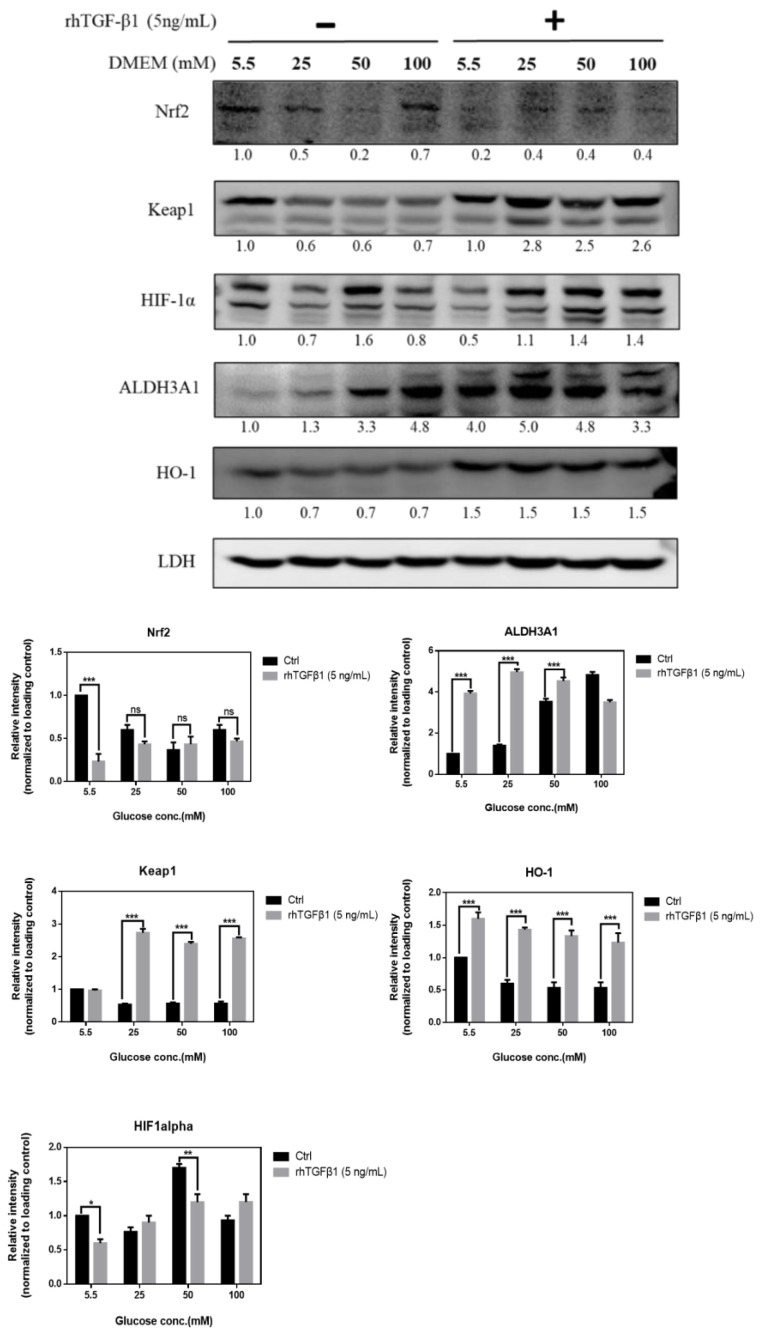
Immunoblot analysis in RGCs with hyperglycemia with or *w*/*o* recombinant TGF-β1 protein (5 ng/mL). Immunoblot analysis of antioxidation pathway-associated proteins (Nrf2, Keap1, HIF-1α, ALDH3A1, and HO-1) under 5.5 mM, 25 mM, 50 mM, and 100 mM glucose concentration medium with or *w*/*o* recombinant TGF-β1 protein. The protein expression values were quantified in relation to the loading control. Data are represented as means ± SEM. *, *p* < 0.05; **, *p* < 0.01; ***, *p* < 0.001; ns, non-significant, when compared to the 5.5 mM group (*n* = 3).

**Figure 8 ijms-21-06482-f008:**
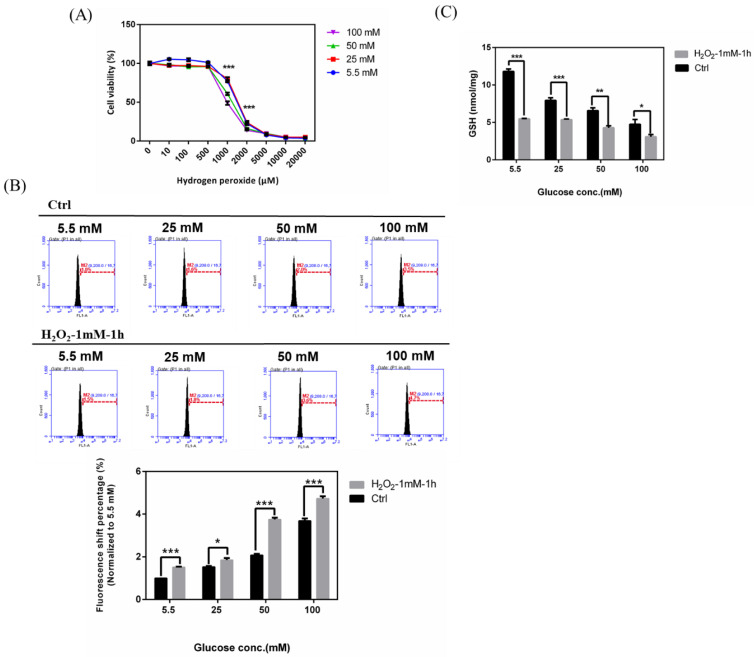
Effects in RGCs with hyperglycemia with or *w*/*o* hydrogen peroxide for 1 h. (**A**) RGC-5 cells were incubated in 5.5 mM, 25 mM, 50 mM, and 100 mM glucose for long-term exposure and seeded into 96-well plates to treat with different concentrations of H_2_O_2_. Cell viability under different concentrations for 1 h was evaluated using the cell viability assay. ***, *p* < 0.001 when compared to the 5.5 mM group. (**B**) Upper: iROS accumulation measurement in RGCs with hyperglycemia with or *w*/*o* hydrogen peroxide (1 mM) for 1 h. Intracellular ROS accumulations were quantified by the APF assay on RGC-5 cell cultures at different glucose concentrations (5.5 mM, 25 mM, 50 mM and 100 mM glucose) with or without hydrogen peroxide (1 mM) for 1 h. After exposure, 106 cells were incubated with APF at a specific concentration (5 μM) in PBS. Intracellular ROS generation was quantified by flow cytometry. Lower: Relative fluorescence shift percentage was quantified and normalized with respect to the Ctrl (5.5 mM). Data are represented as mean ± SEM. *, *p* < 0.05; and ***, *p* < 0.001 when compared to Ctrl. (*n* = 3) (**C**) GSH content in RGCs with hyperglycemia with or *w*/*o* hydrogen peroxide (1 mM) for 1 h. Dose responses involving total GSH contents in RGC-5 cells exposed to 5.5 mM, 25 mM, 50 mM, and 100 mM glucose medium with or *w*/*o* hydrogen peroxide (1 mM) for 1 h were assayed. Data are represented as means ± SEM. *, *p* < 0.05; **, *p* < 0.01; and ***, *p* < 0.001 when compared to the Ctrl.

**Figure 9 ijms-21-06482-f009:**
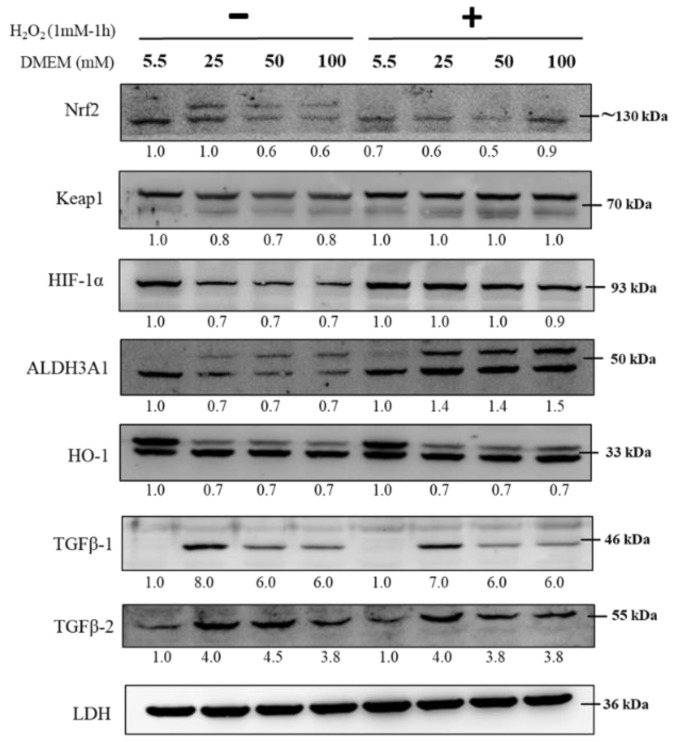
Immunoblot analysis in RGCs with hyperglycemia with or *w*/*o* hydrogen peroxide (1 mM) for 1 h. Immunoblot analysis of antioxidation pathway-associated proteins (Nrf2, Keap1, HIF-1α, ALDH3A1, HO-1, TGFβ-1, and TGFβ-2) in 5.5 mM, 25 mM, 50 mM, and 100 mM glucose supplemented medium with or *w*/*o* hydrogen peroxide (1 mM) for 1 h. The protein expression values were quantified in relation to the loading control. Data are represented as means ± SEM. *, *p* < 0.05; **, *p* < 0.01; ***, *p* < 0.001; ns, non-significant when compared to the 5.5 mM group (*n* = 3).

**Figure 10 ijms-21-06482-f010:**
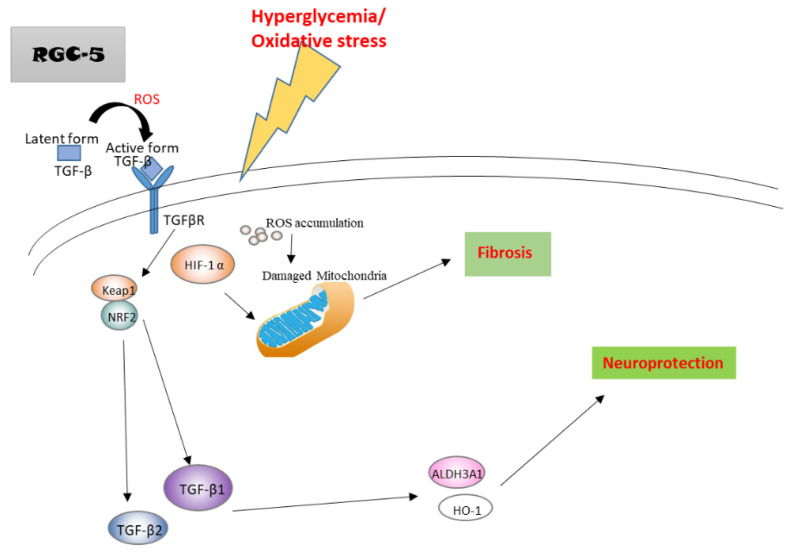
A hypothetical model detailing the role of TGF-β in hyperglycemia.

**Table 1 ijms-21-06482-t001:** Antibodies information.

Antibody	Molecular Weights (kDa)	Company	Monoclonal/Polyclonal	Working Concentration
Nrf2	75/130	Abclonal A1244	rab pAb	1:2000
Keap1	70	Abclonal-A17062	rab pAb	1:2000
HIF-1α	93	Abclonal A11945	rab pAb	1:2000
ALDH3A1	50	Abclonal A13275	rab pAb	1:2000
HO-1	33	GeneTex-GTX101147	rab pAb	1:2000
TGFβ-1	46	Abclonal-A2124	rab pAb	1:2000
TGFβ-2	48/55	Abclonal A3640	rab pAb	1:2000
LDH (Lactate Dehydrogenase)	36	GeneTex-GTX101416	rab pAb	1:2000

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
