# Peer review of "The Role of Transforming Growth Factor-Beta in Retinal Ganglion Cells with Hyperglycemia and Oxidative Stress"

_ijms, 2020, doi:10.3390/ijms21186482_

Round 1
Reviewer 1 Report
In this study, the authors reported the protection effects of Transforming growth factor beta-1 (TGF-1) in retinal ganglion cells grown in a hyperglycemic medium. The protective effects of TGF are well and widely described. Nevertheless, I think that the study is not capable of establishing the pathway the route by which the TGF exerts its effects (the authors only perform western blot and without statistical analysis) but it shed some light on it.
Abstract
Although it is interesting to understand the pathway by which TGF exerts its protective effects, I think that the most consistent results of the study refer to the effects of this protein on retinal ganglion cells.
Introduction
Lines 35-37 - “The underlying cause leading to retinopathy remains unclear, however, the study has implicated that hypoxia, ischemia, oxidative stress, and metabolic mechanisms to be involved in the pathogenesis [1]. Indeed, the order of pathological events in diabetic retinopathy is not fully understood, but the authors should refer to other relevant events such as inflammation and neuroglial degeneration (recent studies indicate that it may precede microvascular changes). There are some examples of more recent bibliography:
1. Lechner J, O’Leary OE, Stitt AW (2017) The pathology associated with diabetic retinopathy. Vision Res 139:7–14. https://doi.org/10.1016/j.visres.2017.04.003
2. Wong TY, Cheung CMG, Larsen M, et al (2016) Diabetic retinopathy. Nat Rev Dis Prim 2:16012. https://doi.org/10.1038/nrdp.2016.12
3. Wang W, Lo ACY (2018) Diabetic retinopathy: Pathophysiology and treatments. Int J Mol Sci 19:. https://doi.org/10.3390/ijms19061816
4. Kusuhara S, Fukushima Y, Ogura S, et al (2018) Pathophysiology of diabetic retinopathy: The old and the new. Diabetes Metab J 42:364–376. https://doi.org/10.4093/dmj.2018.0182
5. Duh EJ, Sun JK, Stitt AW (2017) Diabetic retinopathy: current understanding, mechanisms, and treatment strategies. JCI insight 2:1–13. https://doi.org/10.1172/jci.insight.93751
Lines 37-48 – Please add more citations in this paragraph. I also noticed that some of the citations could be more recent.
Results
Lines 146-147 – “Also, TGF-β1 & TGF-β2 knockdown promotes more thick stress fibers formation in each glucose concentration group.” Do the authors have seen more considerable differences in fibers formation between TGF-β1 & TGF-β2 knockdown and controls with the increasing concentration of glucose? Although the figure has enough quality, some of the described differences are not easily seen.
Lines 175-178 – “Meanwhile, we investigated the changes of GSH content in TGF-β1/2 knockdown RGCs incubated in 5.5, 25, 50, and 100 mM DMEM. We found that the shTGF-β1 and shGF-β2 treated groups showed reduced cellular GSH content in each glucose concentration, especially the shTGF-β1 group. These results also showed that TGF-β1 knockdown aggravates hyperglycemia-induced oxidative damage in RGCs (Figure 4C).” An interesting result is that this effect of knockdown is more significant at low glucose concentrations, while the effects of TGF overexpression are more significant at high glucose concentrations. “We found 230 that the groups with rhTGF-β1 (5 ng/mL) added show the higher cellular GSH content in each glucose concentration, especially in 50 mM and 100 mM. (lines 230-232).” It would be interesting to add this fact to the discussion.
Lines 215-217 – “we established experiments with rhTGF-β1 protein (5ng/mL) followed by detecting the effects of TGF-β upregulation on hyperglycemia-triggered oxidative damage.” How the authors select the concentration of TGF (5ng/mL)? If I understand well it was based in previous experiments but which were the criteria? Also, I understand the selection of rhTGF-β1 protein instead of TGF-β2 protein based on your results but the authors did perform any assays with TGF-β2?
Figures 1,4 and 6 have poor quality. The authors should increase the quality of these figures.
Another concern about this study is the lack of statistical analysis in western blot analysis. The authors didn’t perform enough assays to assess the statistical differences? Although I understand that the authors analyze many proteins by western blot, the statistical analysis should be performed at least in the proteins with more visible differences.
Discussion
Lines 328-331 – “In the previous research, we 328 indicated the important role of TGF-β1/2 in regulating oxidative stress-induced damage in RGCs and 329 showed that TGF-β1/2 knockdown promoted apoptosis and ROS accumulation in RGCs induced by 330 H2O2 treatment.” The authors could cite their previous work.
Lines 387-391 – “When we combined the two oxidative stress-inducible 387 factors, hyperglycemia and hydrogen peroxide, the data revealed that the tolerance to H2O2 in high glucose concentration became lower than ctrl group, these data implied that in patients with diabetes combined with glaucoma state, the damage to RGCs is more serious than those whose plasma glucose level are in control. Additional experiments, however, will be needed to further clarify the significance of signal transduction.” The authors should take into consideration that the oxidative stress caused by hyperglycemia (accumulation of AGEs) is one of the multiple mechanisms that explain the effects of high glucose concentration in the eye and the onset of diabetic retinopathy.
Materials and Methods
Lines 409-412 - How the authors did the correspondence between the glucose concentrations in medium and blood glucose levels?
Author Response
In this study, the authors reported the protection effects of Transforming growth factor beta-1 (TGF-1) in retinal ganglion cells grown in a hyperglycemic medium. The protective effects of TGF are well and widely described. Nevertheless, I think that the study is not capable of establishing the pathway the route by which the TGF exerts its effects (the authors only perform western blot and without statistical analysis) but it shed some light on it.
Abstract
Although it is interesting to understand the pathway by which TGF exerts its protective effects, I think that the most consistent results of the study refer to the effects of this protein on retinal ganglion cells.
Response: Thanks for reviewer’s comments. Indeed, our results showed that the TGF-β mediated enhancement of antioxidant signaling was correlated with the activation of stress response proteins and antioxidant pathway, such as ALDH3A1, HO-1, Nrf2, and HIF-1α; while demonstrated the protective effects that TGF-β keeps RGCs from hyperglycemia-triggered harm. However, additional experiments will be needed to further clarify the significance of pathway and up/down stream of signal transduction.
Introduction
Lines 35-37 - “The underlying cause leading to retinopathy remains unclear, however, the study has implicated that hypoxia, ischemia, oxidative stress, and metabolic mechanisms to be involved in the pathogenesis [1]. Indeed, the order of pathological events in diabetic retinopathy is not fully understood, but the authors should refer to other relevant events such as inflammation and neuroglial degeneration (recent studies indicate that it may precede microvascular changes). There are some examples of more recent bibliography:
- Lechner J, O’Leary OE, Stitt AW (2017) The pathology associated with diabetic retinopathy. Vision Res 139:7–14. https://doi.org/10.1016/j.visres.2017.04.003
- Wong TY, Cheung CMG, Larsen M, et al (2016) Diabetic retinopathy. Nat Rev Dis Prim 2:16012. https://doi.org/10.1038/nrdp.2016.12
- Wang W, Lo ACY (2018) Diabetic retinopathy: Pathophysiology and treatments. Int J Mol Sci 19:. https://doi.org/10.3390/ijms19061816
- Kusuhara S, Fukushima Y, Ogura S, et al (2018) Pathophysiology of diabetic retinopathy: The old and the new. Diabetes Metab J 42:364–376. https://doi.org/10.4093/dmj.2018.0182
- Duh EJ, Sun JK, Stitt AW (2017) Diabetic retinopathy: current understanding, mechanisms, and treatment strategies. JCI insight 2:1–13. https://doi.org/10.1172/jci.insight.93751
Lines 37-48 – Please add more citations in this paragraph. I also noticed that some of the citations could be more recent.
Response: Thanks for reviewer’s suggestion. We have added the statements in the Introduction section and cited above & other important recent references in the manuscript.
Results
Lines 146-147 – “Also, TGF-β1 & TGF-β2 knockdown promotes more thick stress fibers formation in each glucose concentration group.” Do the authors have seen more considerable differences in fibers formation between TGF-β1 & TGF-β2 knockdown and controls with the increasing concentration of glucose? Although the figure has enough quality, some of the described differences are not easily seen.
Response: Thanks for reviewer’s comments to enhance the quality of this manuscript. We have represented these data and showed more significant differences.
Lines 175-178 – “Meanwhile, we investigated the changes of GSH content in TGF-β1/2 knockdown RGCs incubated in 5.5, 25, 50, and 100 mM DMEM. We found that the shTGF-β1 and shTGF-β2 treated groups showed reduced cellular GSH content in each glucose concentration, especially the shTGF-β1 group. These results also showed that TGF-β1 knockdown aggravates hyperglycemia-induced oxidative damage in RGCs (Figure 4C).” An interesting result is that this effect of knockdown is more significant at low glucose concentrations, while the effects of TGF overexpression are more significant at high glucose concentrations. “We found 230 that the groups with rhTGF-β1 (5 ng/mL) added show the higher cellular GSH content in each glucose concentration, especially in 50 mM and 100 mM. (lines 230-232).” It would be interesting to add this fact to the discussion.
Response: Thanks for reviewer’s suggestion. We have stated this point in the Discussion section.
Lines 215-217 – “we established experiments with rhTGF-β1 protein (5ng/mL) followed by detecting the effects of TGF-β upregulation on hyperglycemia-triggered oxidative damage.” How the authors select the concentration of TGF (5ng/mL)? If I understand well it was based in previous experiments but which were the criteria?
Also, I understand the selection of rhTGF-β1 protein instead of TGF-β2 protein based on your results but the authors did perform any assays with TGF-β2?
Response: Thanks for reviewer’s comments. Actually, we have checked a series of references about the concentration of rhTGF-β1 during the process of research. In fact, we did several different concentrations (1, 5, 10, 20 ng/mL) of rhTGF-β1, and finally came to the conclusion that 5 ng/mL rhTGF-β1 protects RGCs against hyperglycemia-triggered oxidative damage. Accordingly, the dose too high or too low is hardly to achieve the protective effects. However, these statement was reasonable when we discussed about RGC, while in other ocular tissue, the conditions will be changed.
The results in shTGF-β2 were not significantly showed the differences when compared to the data in shTGF-β1. To narrow down our results, the further investigation only focused on TGF-β1. This is the reason why the data of shTGF-β2 is absent.
Figures 1,4 and 6 have poor quality. The authors should increase the quality of these figures.
Another concern about this study is the lack of statistical analysis in western blot analysis. The authors didn’t perform enough assays to assess the statistical differences? Although I understand that the authors analyze many proteins by western blot, the statistical analysis should be performed at least in the proteins with more visible differences.
Response: Thanks for reviewer’s suggestion. We have refined these images and added the statistical analysis in the western blot.
Discussion
Lines 328-331 – “In the previous research, we 328 indicated the important role of TGF-β1/2 in regulating oxidative stress-induced damage in RGCs and 329 showed that TGF-β1/2 knockdown promoted apoptosis and ROS accumulation in RGCs induced by 330 H2O2 treatment.” The authors could cite their previous work.
Response: Thanks for reviewer’s suggestion. Our previous work is pending revision and under review.
Lines 387-391 – “When we combined the two oxidative stress-inducible 387 factors, hyperglycemia and hydrogen peroxide, the data revealed that the tolerance to H2O2 in high glucose concentration became lower than ctrl group, these data implied that in patients with diabetes combined with glaucoma state, the damage to RGCs is more serious than those whose plasma glucose level are in control. Additional experiments, however, will be needed to further clarify the significance of signal transduction.” The authors should take into consideration that the oxidative stress caused by hyperglycemia (accumulation of AGEs) is one of the multiple mechanisms that explain the effects of high glucose concentration in the eye and the onset of diabetic retinopathy.
Response: Thanks for reviewer’s comments and suggestions to enhance the quality of this manuscript. We have stated this point in the Discussion section.
Materials and Methods
Lines 409-412 - How the authors did the correspondence between the glucose concentrations in medium and blood glucose levels?
Response: Thanks for reviewer’s comments. The experimental conditions in our study were 5.5, 25, 50, and 100 mM glucose concentration medium, and the corresponding blood glucose level is 100, 454, 910, and 1600 mg/dl, respectively. However, when we discuss the data of the group of 5.5 mM and 25 mM, we can find that the data in the 25 mM group shows better cell function and capacity or viability. This is the reason why 25 mM is the glucose concentration of normal cell line maintaining medium. This also explains why the data from 5.5 mM glucose supplementation is usually not the perfect condition in cell functional assay. Nevertheless, our purpose is to investigate the effects of oxidative stress and its possible signal transduction in the cellular level. While the 25 mM glucose concentration medium is equivalent to 454 mg/dl on blood glucose level, which is the extreme value in diabetes patients, it seems to the the normal state in the in vitro cell culture. Therefore, we established the other two conditions, 50, and 100 mM glucose concentration, which were comparatively extreme hyperglycemia-induced oxidative stress.
Reviewer 2 Report
Introduction section:
The introduction section should be expanded. Please, better describe the pathogenic events triggered by hyperglycemia in diabetic retinopathy.
The authors should include some references such as:
- Toxicol In Vitro. 2017 Oct;44:182-189.
- Biomed Res Int. 2014;2014:801269. doi:10.1155/2014/801269
Results section:
-Figure 2, 5, 7, 9. A very limited part of the immunoblots is shown on the figures and without molecular size markers. Larger part of the blots and the position molecular weight marker bands must be shown. Moreover, the authors have to include the densitometric analyses of western blot signals through bar graphs to better identify differences of protein expression.
-A table describing the antibodies would be useful.
Author Response
Introduction section:
The introduction section should be expanded. Please, better describe the pathogenic events triggered by hyperglycemia in diabetic retinopathy.
The authors should include some references such as:
- Toxicol In Vitro. 2017 Oct;44:182-189.
- Biomed Res Int. 2014;2014:801269. doi:10.1155/2014/801269
Response: Thanks for reviewer’s suggestion. We have added the description in the Introduction section and cited above & other important references in the manuscript.
Results section:
-Figure 2, 5, 7, 9. A very limited part of the immunoblots is shown on the figures and without molecular size markers. Larger part of the blots and the position molecular weight marker bands must be shown. Moreover, the authors have to include the densitometric analyses of western blot signals through bar graphs to better identify differences of protein expression.
Response: Thanks for reviewer’s suggestion. We have represented these images and added the statistical analysis in the western blot.
-A table describing the antibodies would be useful.
Response: Thanks for reviewer’s comments and suggestions to enhance the quality of this manuscript. We have added the antibodies table.
Round 2
Reviewer 2 Report
The authors should include molecular size markers and densitometric analyses in the western blot of Figure 5, 7 and 9.
Author Response
The authors should include molecular size markers and densitometric analyses in the western blot of Figure 5, 7 and 9.
Response: Thanks for reviewer’s suggestion. We have added the molecular size markers and statistical analysis in the western blot of Figure 5, 7 and 9.